# Highly selective and robust single-atom catalyst Ru$_1$/NC for reductive amination of aldehydes/ketones

Haifeng Qi[1,2,4], Ji Yang[1,4], Fei Liu[1,4], LeiLei Zhang ◯ [1✉], Jingyi Yang[1,2], Xiaoyan Liu ◯ [1], Lin Li[1], Yang Su[1], Yuefeng Liu[1], Rui Hao[1], Aiqin Wang ◯ [1,3✉] & Tao Zhang ◯ [1,3✉]

Single-atom catalysts (SACs) have emerged as a frontier in heterogeneous catalysis due to the well-defined active site structure and the maximized metal atom utilization. Nevertheless, the robustness of SACs remains a critical concern for practical applications. Herein, we report a highly active, selective and robust Ru SAC which was synthesized by pyrolysis of ruthenium acetylacetonate and N/C precursors at 900 °C in N$_2$ followed by treatment at 800 °C in NH$_3$. The resultant Ru$_1$-N$_3$ structure exhibits moderate capability for hydrogen activation even in excess NH$_3$, which enables the effective modulation between transimination and hydrogenation activity in the reductive amination of aldehydes/ketones towards primary amines. As a consequence, it shows superior amine productivity, unrivalled resistance against CO and sulfur, and unexpectedly high stability under harsh hydrotreating conditions compared to most SACs and nanocatalysts. This SAC strategy will open an avenue towards the rational design of highly selective and robust catalysts for other demanding transformations.

[1] CAS Key Laboratory of Science and Technology on Applied Catalysis, Dalian Institute of Chemical Physics, Chinese Academy of Sciences, Dalian, China. [2] University of Chinese Academy of Sciences, Beijing, China. [3] State Key Laboratory of Catalysis, Dalian Institute of Chemical Physics, Chinese Academy of Sciences, Dalian, China. [4] These authors contributed equally: Haifeng Qi, Ji Yang, Fei Liu. ✉email: zhangleilei@dicp.ac.cn; aqwang@dicp.ac.cn; taozhang@dicp.ac.cn

Single-atom catalysts (SACs) have emerged as a class of heterogeneous catalysts for they offer the potential to maximize the metal utilization efficiency and the opportunity to probe into the reaction mechanism at the atomic scale in a manner of resembling molecular catalysts[1–5]. In most SAC systems, the metal single atoms are bonded to the support via strong electronic interaction, which renders them to be positively charged and thereby weakens their capability for activating hydrogen molecule[6,7]. Such a weak to moderate ability to activate hydrogen is just what is required for a certain of chemoselective hydrogenation reactions[8], and this has been sufficiently demonstrated in the chemoselective hydrogenation of substituted nitroarenes since the birth of SACs[9–13]. Nevertheless, SACs have not been explored for more challenging reactions which involve both hydrogen and other strongly adsorbed molecules.

The reductive amination of aldehydes/ketones to produce primary amines is such a reaction that involves the activation of both hydrogen and $NH_3$ molecules[14]. In the past few years, this reaction has attracted intensive attention attributed to the broad applications of primary amines as key building blocks for pharmaceuticals, agrochemicals, polyamides, and other fine chemicals[15,16]. In particular, the aldehydes/ketones are readily available from lignocellulosic biomass, which provides the opportunity for the sustainable production of primary amines from renewables and thus contributes to reducing the carbon footprint[17,18]. Nevertheless, the selectivity maneuvering toward primary amines remains a great challenge owing to the occurrence of many side reactions. Taking the reductive amination of furfural (FAL) to furfural amine (FAM) as an example[19], it involves a complex reaction network as shown in Fig. 1. While the target product FAM (2a) can be facilely produced through the condensation of FAL (1a) with $NH_3$ followed by the subsequent hydrogenation of primary imine (7a), there are at least three side reactions to compete with the main reaction due to the high reactivity of aldehyde (1a) as well as the high instability of the intermediate imine (7a): the direct hydrogenation of FAL to furfural alcohol (6a), the trimerization of 7a and the subsequent cyclization to form 3a, and the condensation of 2a with 1a or 7a to form a stable intermediate, Schiff base (secondary imine, 4a), which can be further hydrogenated to secondary amine (5a). This network involving many possible reaction pathways imposes a

great challenge for the selective synthesis of primary amines. Moreover, different from simple hydrogenation, reductive amination requires the catalyst to be able to survive in the presence of excess ammonia and amine, which are usually strongly adsorbed on metal catalysts and become a possible poison for hydrogenation reactions[20].

Based on the above understanding of the reaction mechanism, we propose that a highly selective catalyst for reductive amination should possess weak to moderate adsorption to hydrogen in the presence of $NH_3$. If a catalyst has strong hydrogenation activity such as Pd and Pt, the secondary amine and/or alcohol would be favorably obtained[21]. On the other hand, if a catalyst cannot activate hydrogen, imine trimerization would occur to a great extent[22]. Therefore, to rationally design catalysts for the reductive amination of aldehydes/ketones towards the primary amines, one needs to be able to effectively tune the hydrogenation activity and to establish a structure–performance relationship at an atomic/molecular scale. Unfortunately, for conventional nanocatalysts, the fine-tuning of their hydrogenation capability is hindered by the heterogeneity in shape, size, and composition of the metal nanoparticles[23,24]. Taking Ru nanocatalysts as an example, the oxidation state, size, and shape of Ru nanoparticles were all reported to govern the catalytic performances in reductive amination reaction[19,25–27]. For instance, we previously found that a small proportion of positively charged $RuO_x$ in Ru/$ZrO_2$ catalyst played a critical role in the activation of carbonyl group[25]. Hara's group stated the weak electron donation of Ru in Ru/$Nb_2O_5$ catalyst suppressed the over-hydrogenation of furan ring in the reductive amination of furfuraldehyde[19]. Khodakov et al. proposed the dehydrogenation of primary amines was structure-sensitive such that the selectivity of secondary amines increased with the size of Ru nanoparticle[26]. Chandra et al. found the flat-shaped fcc-Ru nanoparticles exposing mainly {111} facets exhibited much higher catalytic activity[27]. While these catalysts exhibited moderate to high selectivity toward the primary amines, no unambiguous structure–performance relationship has been established, which hinders the rational design of efficient and selective catalysts. In addition to Ru nanocatalysts, it was recently reported that nanoparticles (NPs) of Ni[28,29] or Co[30], when they were encapsulated with an N-doped carbon layer via pyrolysis of organic ligands, exhibited greatly enhanced selectivity to the

**Fig. 1 Reaction network for the reductive amination of furfural.** The transformation of furfural (1a) to the target product furfural amine (2a) is accompanied by a number of side reactions (3a– 8a).

primary amines. Again, due to the heterogeneity in metal size and the complex structure of the N-doped carbon layer, the underlying mechanism is not clear yet.

Thanks to the uniform and tunable structure at the atomic scale, SACs provide a unique platform to probe into the structure–performance relationship[31–34]. We and other research groups have reported the fine modulation of the hydrogenation activities of $Pt_1/FeO_x$[33], Co–N–C[35], and Ni–N–C[36] SACs through engineering the coordination environment of SACs. Herein, we apply SACs to a more demanding reaction, the reductive amination of biomass-derived aldehydes/ketones for the selective synthesis of primary amines. $Ru_1/NC$ SACs with finely tuned active sites including $RuN_5$, $RuN_4$, and $RuN_3$ are successfully fabricated by changing the synthesis temperatures. It is found that the oxidation state of Ru single atoms is +2, yet the electron density increases with a decrease in the Ru–N coordination number, which results in a gradual enhancement in the hydrogen activation even in the case of strong pre-adsorption of $NH_3$. As a consequence, the selectivity to primary amines is greatly enhanced up to 99%. More interesting, compared with conventional Ru/AC as well as other state-of-the-art Ru nanocatalysts, the $Ru_1/NC$ SAC with $RuN_3$ structure not only possesses the highest primary amine productivity per atom Ru per hour but also exhibits far superior resistance against being poisoned by sulfur and CO, as well as much higher tolerance to harsh treatment including hydrogen reduction at 600 °C. To our knowledge, the as-obtained $RuN_3$ moiety is the most robust yet highly active and selective single-atom structure reported thus far.

## Results and discussion

**Preparation and textual properties of $Ru_1/NC$ SACs.** $Ru_1/NC$ SACs were prepared by ball milling of a mixture of L-cysteine, dicyandiamide, and ruthenium acetylacetonate $(Ru(acac)_3)$, followed by pyrolysis under $N_2$ atmosphere (Fig. 2a), similar to that earlier reported by Liu et al. for the synthesis of Ni/NC[37]. During the pyrolysis process, the C, N-containing L-cysteine, and dicyandiamide underwent polymerization and condensation reactions to form an N-doped carbon matrix, which then captured the Ru atoms to form $RuN_x$ moieties. With the attempt to tailor the Ru–N coordination environment, the samples were pyrolyzed at different temperatures in the range of 700–1000 °C, and denoted as $Ru_1/NC-T$ (T refers to the pyrolysis temperature). In addition, the $Ru_1/NC-900$ sample was subjected to further treatment by $NH_3$ at 800 °C (denoted as $Ru_1/NC-900–800NH_3$) in order to increase the accessibility of $RuN_x$ active sites through etching of carbonaceous materials by $NH_3$. The as-synthesized $Ru_1/NC-T$ samples show two-dimensional (2D) few-layer graphene morphology (Fig. 2b) with the specific surface area of 216–286 $m^2/g$ and mesoporous structure (Supplementary Fig. 1 and Supplementary Table 1). After $NH_3$ treatment, the specific surface area is significantly increased from 216 to 471 $m^2/g$ due to the creation of plenty of micropores. Inductively coupled plasma optical emission spectrometry (ICP-OES) analysis (Supplementary Table 1) reveals that the mass loading of Ru increases steadily from 0.8 wt% to 1.6 wt% with the pyrolysis temperature, and rises further to 2.3 wt% upon $NH_3$ treatment, suggesting the gradual decomposition of the N-doped carbon matrix at elevated temperatures, especially in the presence of $NH_3$.

The X-ray diffraction (XRD) patterns of all the above samples do not show any reflections of metallic or oxidic Ru species (Supplementary Fig. 2), which, in combination with the absence of NPs in low-magnification scanning transmission electron microscopy (STEM) images (Supplementary Fig. 3), indicates that Ru species are highly dispersed in the N-doped carbon matrix. We then employed sub-Ångström-resolution high-angle annular dark-field scanning transmission electron microscopy (HAADF-STEM) technique to probe the highly dispersed Ru-containing species. Representative images are shown in Fig. 2c, d, and more images are shown in Supplementary Fig. 4. It can be clearly seen that a high density of Ru single atoms (bright dots) are uniformly

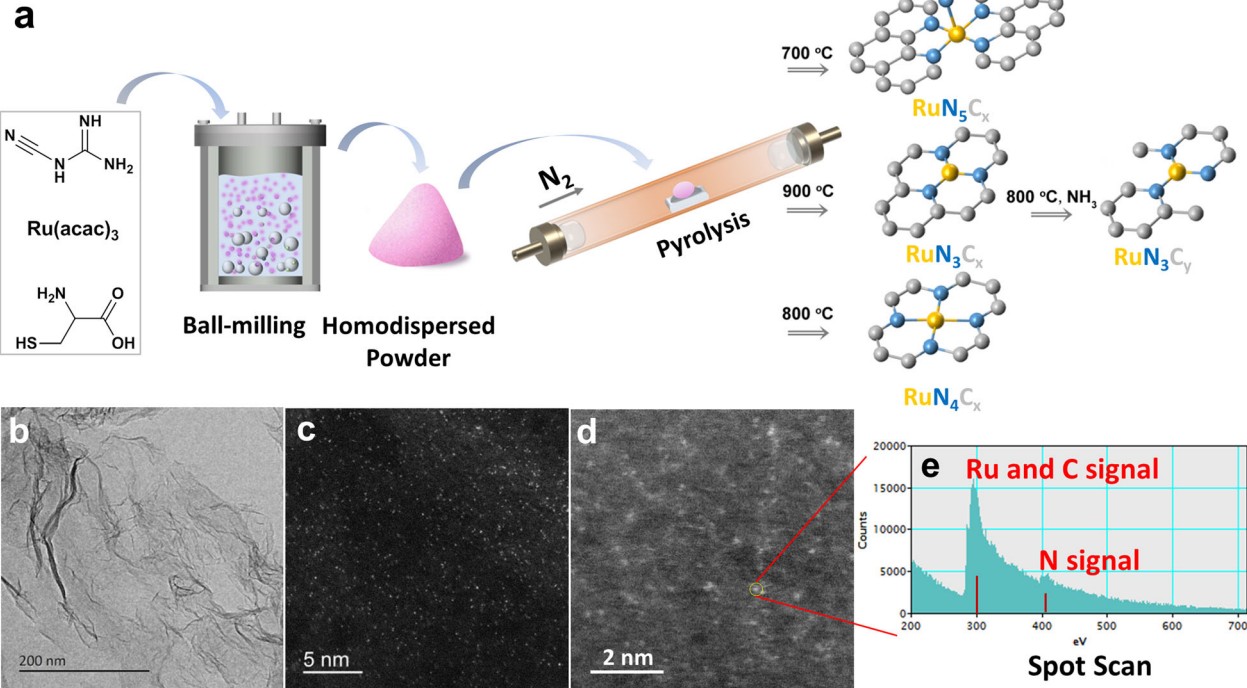

**Fig. 2 Preparation and characterization for $Ru_1/NC-T$ catalysts. a** Schematic illustration for preparation of $Ru_1/NC-T$ catalysts. **b** Transmission electron microscopy image of $Ru_1/NC-900$. **c, d** High-angle annular dark-field scanning transmission electron microscopy images of $Ru_1/NC-900$ and $Ru_1/NC-900–800NH_3$. **e** Electron energy loss spectrometer spectra of $Ru_1/NC-900–800NH_3$.

dispersed on the N-doped carbon materials. The surface density of single atoms based on Ru loading and the specific surface area of the Ru$_1$/NC-900 is calculated to be 0.5 Ru atom/nm$^2$, which is lower than that estimated from HAADF-STEM images (1.2 Ru atom/nm$^2$), possibly due to that some Ru single atoms were imbedded into the NC material. Notably, NH$_3$ treatment of Ru$_1$/NC-900 at 800 °C does not change the atomic dispersion of Ru species, manifesting the effectiveness of our method to prepare Ru SACs. It should be emphasized that the acid-leaching step widely employed to remove the metal nanoparticles in the preparation of Co(Fe, Ni)–N–C single-atom catalysts[35,36,38] is absolutely unnecessary in our method because all of the Ru species are atomically dispersed, which is highly desirable from the viewpoint of saving precious metals and environmentally benign process. When the sample was pyrolyzed at 1000 °C, however, serious aggregation began to take place leading to the formation of a large number of Ru NPs with the average size of 5.2 nm (Supplementary Fig. 5). Obviously, in order to achieve stable Ru single atoms on the NC surface, the pyrolysis temperature should not exceed 1000 °C. The microenvironment of Ru single atoms is further probed with high-resolution energy-dispersive spectrometer (EDS) and electron energy loss spectrometer (EELS). As shown in Fig. 2e, both Ru (note, Ru and C EELS spectra are overlapped and therefore cannot be distinguished) and N spectra are detected when the electron beam is focused on an individual Ru single atom, which suggests that Ru is bonded to N in accordance with that reported in M–N–C catalysts[37,39]. The elemental mapping by EDS (Supplementary Fig. 6) shows that Ru and N elements are uniformly dispersed throughout the whole sample, again suggesting that they are bonding together.

**Probing into the electronic and coordinative structure**. X-ray photoelectron spectroscopy (XPS) and X-ray absorption spectroscopy (XAS) characterizations were conducted to determine the chemical state and coordination environment of Ru$_1$/NC-T catalysts. Due to the overlap in binding energies of Ru 3d and C 1s (284.6 eV), we detected Ru 3p XPS in spite of some loss in signal/noise ratio, as shown in Fig. 3. It is found that the Ru 3p$_{3/2}$ binding energy is in the range of 462.0–462.6 eV, indicating that Ru single atoms have an oxidation state of +2[40]. Moreover, a slight shift toward lower energy can be observed with an increase of the pyrolysis temperature, in particular for the Ru$_1$/NC-900–800NH$_3$ catalyst, indicating a slight increase in the electron density of Ru single atoms. Such a fine-tuning of the electronic property without altering the single-atom dispersion is highly

important to SACs since it can allow for the de-coupling of activity and chemoselectivity on the condition that the latter is more dependent on the single-atom dispersion[31,41,42]. On the other hand, the N 1s XP spectra can be deconvoluted into five different N species: pyridinic N (397.9–398.4 eV), Ru–N (399.0–399.5 eV), pyrrolic N (400.5 eV), graphitic N (401.3–401.6 eV) and oxidized graphitic N (402.5–403.3 eV). It is interesting to note that the binding energies of the Ru–N species also slightly shift toward lower values with the increase of the pyrolysis temperature, suggesting the increased electron density on the N atoms, which is likely caused by electron donation of the neighboring C[43]. Moreover, with an increase of pyrolysis temperature, the contents of both pyridinic N and Ru–N decrease gradually while the other three types of N species increase (Supplementary Table 2), suggesting that Ru atoms are likely coordinated to pyridinic N and the resultant Ru–N coordination number decreases by elevating the pyrolysis temperature and further treatment with NH$_3$.

The modulation of the electronic and coordinative structures is further supported by XAS technique. Figure 4a displays the X-ray absorption near-edge spectra (XANES) at Ru K-edge of Ru$_1$/NC-T catalysts and references. The edge energies ($E_0$) for all the Ru$_1$/NC-T samples are lower than that of Ru(acac)$_3$ yet higher than that of Ru foil, suggesting Ru atoms carry positive charges +δ (0 < δ < 3), consistent with the XPS characterization results. Meanwhile, a comparison of the $E_0$ values suggests the oxidation state increases following the order of Ru$_1$/NC-900–800NH$_3$ < Ru$_1$/NC-900 < Ru$_1$/NC-800 ≈ Ru$_1$/NC-700, again in line with the XPS result. The coordination environment of Ru single atoms is determined by the extended X-ray absorption fine structure spectra (EXAFS). As shown in the Fourier-transformed $k^2$-weighted EXAFS spectra at the Ru K-edge (Fig. 4b), in contrast to the reference samples of Ru foil and RuO$_2$, the Ru$_1$/NC-T catalysts do not show any prominent peaks at the positions of either Ru–Ru shell (2.4 Å) or Ru–O–Ru (3.2) shell, excluding the existence of metallic Ru or RuO$_x$ nanoparticles. The absence of Ru–Ru scattering is further demonstrated by the wavelet transform (WT) technique. As shown in Fig. 4c, Ru foil affords a lobe at (2.4 Å, 9.2 Å$^{-1}$), which is attributed to Ru–Ru coordination. The absence of lobe at high k value in Ru$_1$/NC-900–800NH$_3$ (Fig. 4d) indicates the central Ru does not bind to heavy atoms (e.g., Ru), on the contrary, there exists a lobe at low k values (1.5 Å, 3.5 Å$^{-1}$), which can be ascribed to Ru–N coordination. Notably, in the series of Ru$_1$/NC-T samples, the peak intensity of Ru–N contribution decreases with an increase of pyrolysis

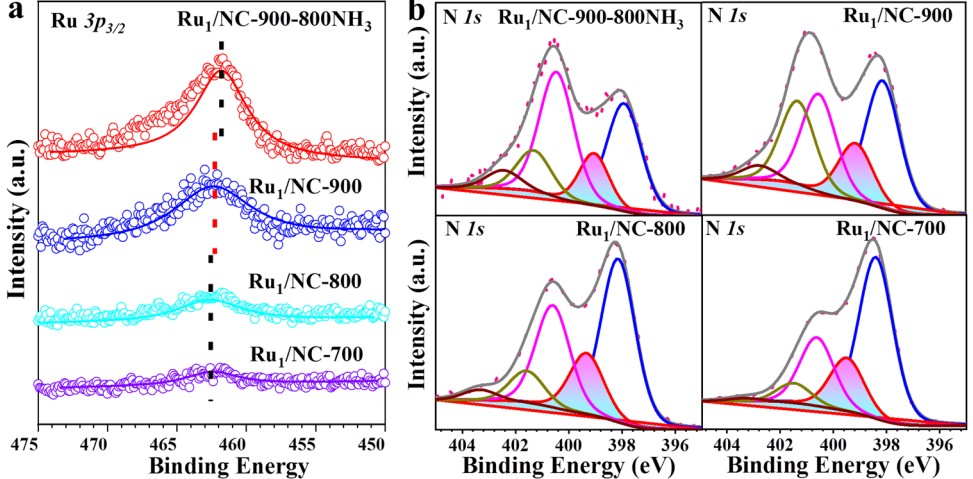

**Fig. 3 X-ray photoelectron spectroscopy of Ru$_1$/NC-T samples. a** Ru 3p$_{3/2}$. **b** N 1s.

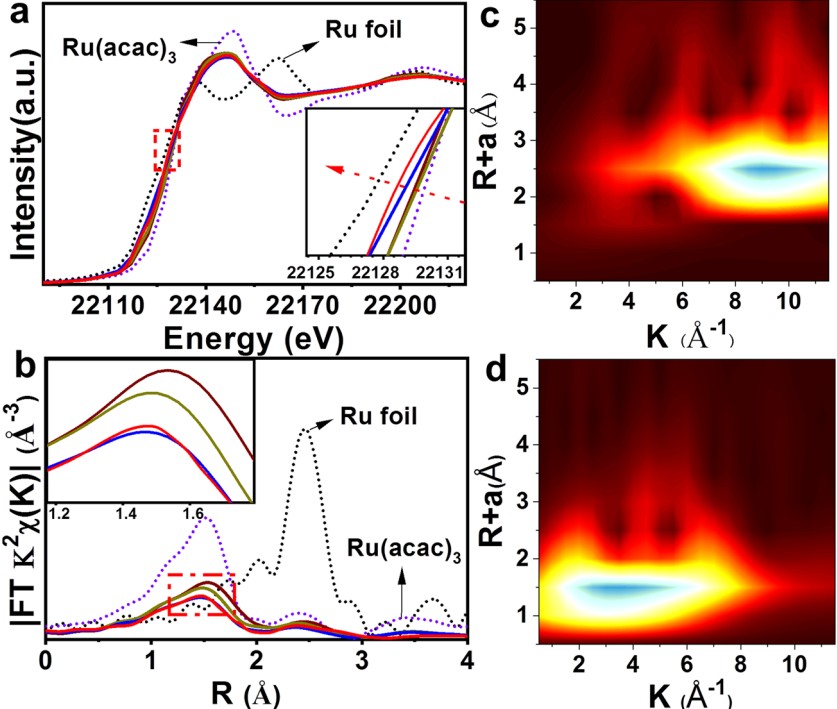

**Fig. 4 X-ray absorption spectra of Ru₁/NC-T catalysts.** **a** The normalized X-ray absorption near-edge spectra at the Ru $K$-edge. **b** The $k^2$-weighted Fourier transform extended X-ray absorption fine structure spectra (EXAFS) in $r$-space. **c**, **d** Wavelet Transformation for the $k^2$-weighted EXAFS signal of Ru foil and Ru₁/NC-900-800NH₃ sample. Red line: Ru₁/NC-900-800NH₃, Blue: Ru₁/NC-900, Dark yellow: Ru₁/NC-800, Wine red: Ru₁/NC-700.

**Table 1 The best-fitted EXAFS results for Ru₁/NC-T catalysts[a].**

| Sample | Shell | CN | $R$ (Å) | $\sigma^2$ ($10^{-2}$ Å²) | $\Delta E_0$ (eV) | $r$-factor (%) |
|---|---|---|---|---|---|---|
| Ru foil | Ru–Ru | 12 | 2.67 | 0.4 | 6.2 | 1 |
| Ru₁/NC-700 | Ru–N | 4.9 | 2.09 | 0.6 | −0.7 | 0.7 |
| Ru₁/NC-800 | Ru–N | 4.3 | 2.05 | 0.5 | −4.0 | 0.2 |
| Ru₁/NC-900 | Ru–N | 3.1 | 2.04 | 0.5 | −3.8 | 0.4 |
| Ru₁/NC-900-800NH₃ | Ru–N | 3.3 | 2.03 | 0.5 | −0.6 | 0.9 |

[a]CN is the coordination number for the absorber–backscatterer pair, $R$ is the average absorber–backscatterer distance, $\sigma^2$ is the Debye-Waller factor, and $\Delta E_0$ is the inner potential correction. The accuracies of the above parameters are estimated as CN, ±20%; R, ±1%; $\sigma^2$, ±20%; $\Delta E_0$, ±20%. The data range used for data fitting in $k$-space ($\Delta k$) and $R$-space ($\Delta R$) are 3.0–12 Å⁻¹ and 1.2–3.2 Å, respectively.

temperature (Fig. 4b, inset), suggesting the decrease of Ru–N coordination number. Exactly, the EXAFS data-fitting results (Table 1) show that the Ru–N coordination number gradually decreases from 4.9 in Ru₁/NC-700 to 4.3 in Ru₁/NC-800 and further to 3.1–3.3 in Ru₁/NC-900 and Ru₁/NC-900–800NH₃, respectively, demonstrating RuN₅, RuN₄, and RuN₃ entities are selectively fabricated by varying the pyrolysis temperature. In addition, the Ru–N distance also continuously decreases from 2.09 Å in Ru₁/NC-700 to 2.03 Å in Ru₁/NC-900–800NH₃, indicating the enhanced Ru–N coordination strength.

**Catalytic performance of Ru₁/NC-T catalysts in reductive amination reaction.** The different coordination environments (i.e., RuN₅, RuN₄, and RuN₃) of Ru single atoms will lead to significant differences in catalytic performances. Here, we choose the reductive amination of furfural (FAL) as the probe reaction considering that FAL is readily available from biomass and the target product furfuryl amine (FAM) is widely used in the manufacture of pharmaceuticals and pesticides[19,44]. It should be noted that neither Ru(acac)₃ nor the mixture of Ru(acac)₃ and N, C-precursor gives the target FAM; instead, oligomers arising from trimerization of imines are produced as the main product. Similar

to the catalyst precursors, the Ru₁/NC-700 does not provide any useful product other than oligomers (Table 2, entry 1), suggesting the negligible hydrogenation activity of the RuN₅ structure. By contrast, with a decrease of the Ru–N coordination number, the target product FAM becomes the predominant one. For example, 43% yield of FAM is achieved on the Ru₁/NC-800 catalyst having a RuN₄ structure (Table 2, entry 2), whereas it is enhanced to 84% over the Ru₁/NC-900 with a RuN₃ structure (Table 2, entry 3). Notably, when the Ru₁/NC-900 is further treated under NH₃ atmosphere, the resultant Ru₁/NC-900–800NH₃ catalyst affords FAM yield as high as 97% (Table 2, entry 4), corresponding to a production rate of 170.7 g_FAM·g_Ru⁻¹ h⁻¹. In contrast to the high efficiency of Ru₁/NC-900 and Ru₁/NC-900–800NH₃ SACs, the Ru₁/NC-1000 catalyst with predominance of Ru NPs provides FAM yield of only 53% (Table 2, entry 5), far inferior to the Ru single-atom catalysts. We also prepared other nanocatalysts including Ru/AC, Ru/Nb₂O₅, and Ru/HZSM-5 which were reported to be highly active for the reductive amination of furfural[19,45,46]. Under identical reaction conditions, the Ru/AC nanocatalyst gives an enhanced FAM yield (82%, Table 2, entry 6) than Ru₁/NC-1000, which can be due to the smaller and more exposed Ru particles on the AC support (Supplementary Fig. 7),

**Table 2 Catalytic performances of $Ru_1$/NC-T single-atom catalysts (SACs) as well as reference Ru nanocatalysts for the reductive amination of furfuraldehyde[a].**

| Entry | Catalysts | Yield (%)[d] | | | Production rate ($g_{FAM}·g_{Ru}^{-1}h^{-1}$) |
|---|---|---|---|---|---|
| | | **2a** | **3a** | **Others** | |
| 1 | $Ru_1$/NC-700 | n.d. | 33 | 67 | – |
| 2 | $Ru_1$/NC-800 | 43 | 9 | 48 | 16.7 |
| 3 | $Ru_1$/NC-900 | 84 | 9 | 7 | 81.5 |
| 4 | $Ru_1$/NC-900-800$NH_3$ | 97 (94[c]) | 3 | n.d. | 170.7 |
| 5 | RuNP-1000 | 53 | 12 | 33 | 20.6 |
| 6 | Ru/AC | 82 | 12 | 6 | 79.5 |
| 7 | Ru/$Nb_2O_5$ | 91 | 3 | 6 | 115.0 |
| 8 | Ru/HZSM-5 | 68 | 12 | 20 | 89.6 |
| 9[b] | $Ru_1$/NC/$Nb_2O_5$ | 97 | n.d. | 3 | 227.6 |

[a]Reaction condition: 2 mmol furfural, catalysts amounts were varied to maintain the molar ratio of Ru:furfural = 1:400, 3 g methanol, 0.5 MPa $NH_3$, 2 MPa $H_2$, 100 °C, 10 h, dodecane as an internal standard. The conversion of FAL in all experiments was>99%.
[b]3 h.
[c]Isolated yield in parentheses.
[d]Others are oligomers; n.d.: not detected.

yet lower than that over the $Ru_1$/NC-900-800$NH_3$. Similarly, both the Ru/$Nb_2O_5$ and Ru/HZSM-5 nanocatalysts deliver an inferior yield of FAM to the $Ru_1$/NC-900-800$NH_3$ SAC even with extended reaction time (Table 2, entries 7–8). To further demonstrate the superiority of the $Ru_1$/NC SAC, we synthesized the $Ru_1$/NC SAC on the $Nb_2O_5$ support given that the electronic interaction between Ru and acidic and reducible $Nb_2O_5$ support would facilitate the reductive amination reaction[19,25]. To our delight, the as-prepared $Ru_1$/NC/$Nb_2O_5$ SAC affords 97% yield in a shorter time of 3 h (Table 2, entry 9), resulting in a production rate of 227.6 $g_{FAM}·g_{Ru}^{-1}h^{-1}$, almost twofold higher than that on the Ru/$Nb_2O_5$ nanocatalyst. These strictly control experiments, together with the comparison with more heterogeneous and homogeneous Ru catalysts reported in the literature (Supplementary Table 3), unequivocally demonstrate the remarkable advantage of $Ru_1$/NC-900–800$NH_3$ SAC in maximizing the atom efficiency of Ru.

Not only possessing a higher production rate than the NPs counterparts but our $Ru_1$/NC-900–800$NH_3$ SAC also exhibits superior stability and resistance against being poisoned by CO or sulfur species. Figure 5a shows that the catalyst can be reused at least five times without any decay in catalytic performance. HAADF-STEM and XAS characterizations reveal that Ru single atoms and the local coordination environment remain unchanged after the reuse tests (Supplementary Fig. 8), demonstrating the excellent stability of the $Ru_1$/NC-900–800$NH_3$ SAC. Figure 5b compares the performances of $Ru_1$/NC-900–800$NH_3$ SAC and Ru/AC nanocatalyst in exposure to strongly adsorbed molecules such as CO and sulfur-containing species. When 1% CO is present in $H_2$ gas, which is the common case when hydrogen comes from reforming natural gas[47], the FAM yield over the $Ru_1$/NC-900–800$NH_3$ SAC goes from 97% slightly down to 88%, a <10% decrease; by contrast, it drastically goes down from 82 to 29% over the Ru/AC nanocatalyst, a more than 50% decrease. Sulfur species, which are frequently present in biomass as well as the biomass-derived platform compounds such as FAL, is another well-known poison to noble metal catalysts[48]. Here we use thiophene as a representative sulfur species to test the catalyst. We are delighted to see that the $Ru_1$/NC-900–800$NH_3$ SAC

manifests high resistance against the sulfur poison; the FAM yield decreases by 18% in the presence of 500 ppm thiophene, which is in stark contrast to a decrease of 67% over the Ru/AC nanocatalyst. Similar to the Ru/AC, the Ru/$Nb_2O_5$ nanocatalyst also shows vulnerability towards being poisoned by CO and sulfur, which is in contrast to the robustness of the $Ru_1$/NC-900–800$NH_3$ SAC (Supplementary Fig. 9). Obviously, the $Ru_1$/NC-900–800$NH_3$ SAC is much superior to typical Ru nanocatalysts in terms of poison-resistance. The ultrarobustness of the $Ru_1$/NC-900–800$NH_3$ SAC is also demonstrated by high-temperature reduction treatment which has been reported to cause serious aggregation of both single atoms and nanoparticles[49–52]. As shown in Supplementary Fig. 10, NPs of Ru are observed only scarcely on the $Ru_1$/NC-900–800$NH_3$ SAC after $H_2$ treatment at 600 °C, which is supported by the absence of Ru–Ru bonding in the EXAFS spectra, indicating the high resistance of Ru single atoms to aggregation. By contrast, serious aggregation takes place after the hydrogen treatment of Ru/AC catalyst, thus significantly reduces the number of active sites. The reaction test reveals that there is a 12% decrease in the FAM yield after the high-temperature hydrogen treatment, which is far less appreciable than the 49% yield decrease over the Ru/AC catalyst (Fig. 5b).

The $Ru_1$/NC-900–800$NH_3$ SAC can be applied to the reductive amination of a broad spectrum of substrates, and good to excellent yields of the corresponding primary amines are obtained. As shown in Table 3, for biomass-derived 5-methyl furfural and 5-hydroxymethyl furfural (Table 3, entries 1–2), high yields (93–98%) of the primary amines are reached. Even for the more challenging substrate, biomass-derived 2,5-diformylfuran, the desired diamine product 2,5-bis(aminomethyl)furan (BAMF) is still obtained with a yield of 63% (Table 3, entry 3), surpassing the earlier reported Raney Ni catalyst[53]. For the aromatic aldehydes with electron-donating/withdrawing groups (Table 3, entries 4–17), the yields of the primary amines are all above 80%. Sensitive functional groups including methoxyl, halides, esters as well as challenging amide and sulfo group are well tolerated. In particular, for heterocyclic aldehydes such as pyridinecarboxaldehyde and 2-thenaldehyde which are well-known poisons to

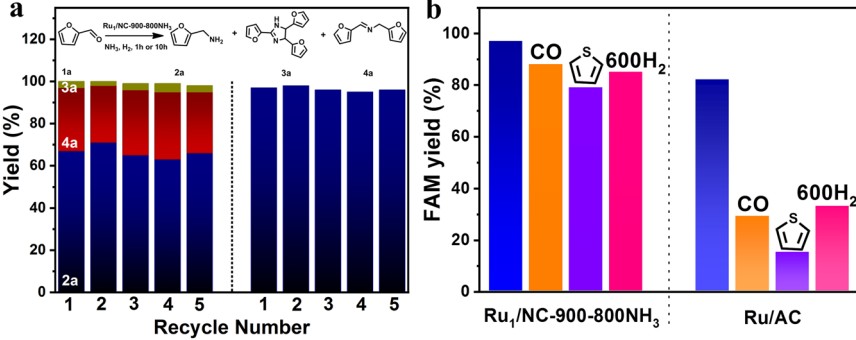

**Fig. 5 Catalyst stability of $Ru_1/NC-900-800NH_3$. a** Reusability tests at different conversion levels. **b** Resistance against CO, sulfur and $H_2$ treatment of $Ru_1/NC-900–800NH_3$ and Ru/AC. Reaction conditions: 2 mmol furfural, 22 mg $Ru_1/NC-900–800NH_3$ or Ru/AC catalysts (0.25 mol% Ru, molar ratio of Ru:furfural = 1:400), 3 g methanol, 0.5 MPa $NH_3$, 2 MPa $H_2$, 100 °C. For reusability tests, data were taken at 1 h (left) and 10 h (right), respectively. For CO poisoning experiment, 2 MPa 1 vol% CO in $H_2$ was used in place of pure $H_2$; for sulfur-poisoning experiment, 500 ppm thiophene was added to the reaction mixture; for $H_2$ treatment test, the catalysts were pre-reduced at 600 °C for 2 h before the reaction.

noble metal catalysts, the reductive amination can still proceed smoothly (Table 3, entries 18–19). Aliphatic aldehydes with or without hydroxyl groups are also good substrates (Table 3, entries 20–23), although a mixture of primary amine and secondary amine is produced for hexaldehyde (Table 3, entry 23). Notably, the reductive amination of the more demanding molecules such as ketones generally requires a more active catalyst or more drastic conditions. Nevertheless, we are gratified to see that the more challenging ketones with various functional groups could also be smoothly converted to the corresponding amines (Table 3, entries 24–35). Finally, the kinetically more sluggish alcohols such as furfural alcohol could be transformed into FAM with a moderate yield at relatively harsh conditions (Table 3, entry 36). All these results demonstrated that the $Ru_1/NC-900–800NH_3$ SAC possessed high activity and selectivity, unrivalled robustness against various poisons and harsh hydrogen treating, as well as excellent tolerance to a wide spectrum of substrates, which is of great potential for practical applications.

**Establishing the structure–performance relationship.** In situ [1]H nuclear magnetic resonance ([1]HNMR) experiments, kinetic studies, and microcalorimetric measurements were performed to get insight into the structure–performance relationship of the series of $Ru_1/NC-T$ catalysts. First, the in- situ [1]HNMR detected the formation of primary imine upon condensation of FAL with $NH_3$ in the absence of $H_2$, while in the presence of $H_2$, both Schiff base (i.e., secondary imine) and FAM were produced associated with the disappearance of primary imine (Supplementary Fig. 11), suggesting both of them come from the hydrogenation of highly unstable intermediate primary imine. Second, conversion-time kinetic profile (Supplementary Fig. 12) reveals that at the beginning stage of the reaction, a large proportion of Schiff base (i.e., secondary imine) was produced, which was then followed by the gradual increase of FAM at the expense of Schiff base. This result indicates that Schiff base is the key intermediate during the reductive amination of FAL toward FAM. As Schiff base is yielded from the condensation between primary amine and aldehyde (or primary imine), its rapid accumulation indicates at least the hydrogenation of primary imine to yield primary amine must proceed fast and therefore cannot be a rate-determining step. Instead, the conversion-time profile shows that after the Schiff base yield reaches the maximum, its subsequent conversion to primary amine proceeds slowly, suggesting the transimination of Schiff base being most likely a rate-determining step in the reductive amination, as shown in Fig. 1.

According to the above reaction mechanism, it is obvious that the catalytic performances of $Ru_1/NC-T$ SACs are governed by their transimination and hydrogenation capabilities. To probe into the adsorption behavior of $NH_3$ and $H_2$ on $Ru_1/NC-T$ catalysts with different coordination environments (i.e., $RuN_5$, $RuN_4$, and $RuN_3$), microcalorimetric measurements were conducted. As shown in Fig. 6a, all the catalysts have strong adsorption to $NH_3$ with initial adsorption heat of ~160 kJ/mol and appear not sensitive to the coordination structure of Ru. By contrast, the adsorption of $H_2$ is highly dependent on the coordinative structure, and both of the initial adsorption heat and uptake increase with a decrease of Ru–N coordination number (Fig. 6a and Supplementary Table 4). Moreover, the $Ru_1/NC-900–800NH_3$ with the same Ru–N CN as the $Ru_1/NC-900$ according to our EXAFS fitting result (Table 1) exhibits much stronger adsorption for $H_2$ than the latter, suggesting that the electronic structure, in addition to geometric structure, significantly affects the adsorption properties of Ru single atoms. Based on XANES and XPS results, the Ru single atoms in the $Ru_1/NC-900–800NH_3$ catalyst are more electron-rich than those in the $Ru_1/NC-900$ structure, which facilitates the electron transfer to the antibonding orbital of $H_2$ molecules and thus activate hydrogen. In our previous work[33], $Pt_1/Fe_2O_3$ SACs with different Pt–O CNs were fabricated and a similar trend was found: the lower the Pt–O CN is, the higher the electron density of Pt single atoms gets, and the higher hydrogenation activity is.

As both $NH_3$ and $H_2$ participate in the reductive amination reaction, we also investigate the adsorption of $H_2$ after pre-adsorption of $NH_3$. In agreement with that reported in other catalyst systems[21], the adsorption of $H_2$ is suppressed to a great extent upon pre-adsorption of $NH_3$ (Fig. 6b and Supplementary Table 4), indicating the competitive adsorption of the two reactants on the active sites. Nevertheless, the suppressing degree is quite different depending on the coordination and electronic structure of Ru single atoms. For $Ru_1/NC-700$ catalyst with the $RuN_5$ structure, the hydrogen adsorption is totally prevented by $NH_3$ adsorption, which may account for its inactivity in the reductive amination reaction. By contrast, hydrogen molecules can still be activated on the other three catalysts with measurable uptakes, and the adsorption strength and uptakes increase in the order of $Ru_1/NC-800 < Ru_1/NC-900 < Ru_1/NC-900-800NH_3$. This experimental result is consistent with the density functional theory (DFT) calculations reported by Zhou et al.[54], which reveals that the smaller the difference between the adsorption energy of $NH_3$ and $H_2$ is, the higher the activity for reductive amination is. Indeed, when the FAM yield is correlated with the $RuN_x$

**Table 3 Production of primary amines from various aldehydes and ketones over $Ru_1$/NC-900-800$NH_3$ catalyst[a].**

| Entry | Substrate | Product | Yield (%) |
|---|---|---|---|
| 1 | | | 98 |
| 2 | | | 93 |
| 3[b] | | | 63 |
| 4 | | | 97 |
| 5 | | | 90 |
| 6 | | | 99 |
| 7 | | | 99 |
| 8 | | | 93 |
| 9 | | | 99 |
| 10[c] | | | 91 (87[h]) |
| 11[c] | | | 81 |
| 12 | | | 95 (90[h]) |
| 13 | | | 91 (88[h]) |
| 14 | | | 91 |
| 15[d] | | | 85 |
| 16 | | | 92 (90[h]) |
| 17 | | | 93 |
| 18 | | | 95 |
| 19[d] | | | 67 |
| 20[e] | | | 81 |
| 21[e] | | | 90 |
| 22 | | | 93 |
| 23 | | | 67/28 |
| 24[f] | | | 95 |
| 25[f] | | | 96 (91[h]) |
| 26[f] | | | 91 |
| 27[f] | | | 81 |
| 28[f] | | | 94 (90[h]) |
| 29[f] | | | 89 (85[h]) |
| 30[f] | | | 73 |
| 31[f] | | | 71 |
| 32[f] | | | 65 |
| 33 | | | 96 |
| 34 | | | 65 |
| 35 | | | 91 |
| 36[g] | | | 61 |

[a]Reaction condition: 2 mmol substrate, 22 mg $Ru_1$/NC-900-800$NH_3$ catalyst, 0.25 mol% Ru, molar ratio of Ru:subatrate = 1:400, 3 g methanol, 0.5 MPa $NH_3$, 2 MPa $H_2$, 100 °C, 10 h, dodecane as an internal standard.
[b]Adding 6 mmol butylamine.
[c]80 °C.
[d]44 mg $Ru_1$/NC-900-800$NH_3$ catalyst, 0.5 mol% Ru, molar ratio of Ru:substrate = 1:200, 120 °C.
[e]3 g 25 wt% aqueous ammonia as solvent, 2 MPa $H_2$, 80 °C, 10 h.
[f]1 MPa $NH_3$.
[g]5 g p-xylene as solvent, 0.8 MPa $NH_3$, 0.2 MPa $H_2$, 180 °C, 20 h.
[h]Isolated yields in parentheses for some selected amine products and NMR and HRMS spectra are shown in Supplementary Figs. 18–24.

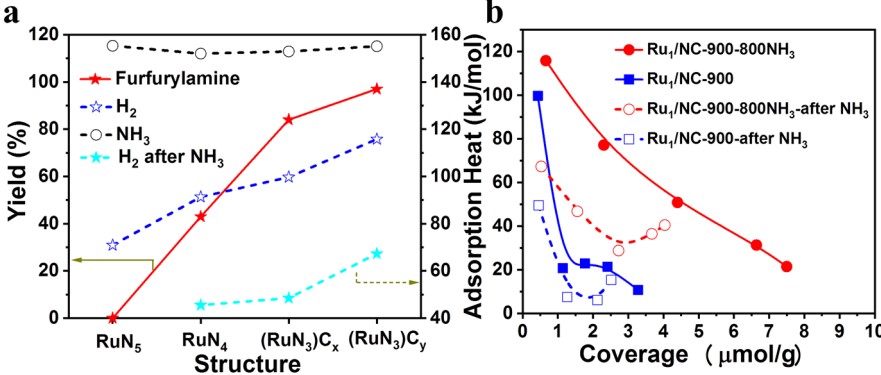

**Fig. 6 Microcalorimetric adsorption of $NH_3$ and $H_2$ on $Ru_1$/NC-T catalysts. a** Initial adsorption heat of $NH_3$ as well as $H_2$ before and after $NH_3$ pre-adsorption as a function of the Ru–N coordination structure. **b** Differential adsorption heat as a function of coverage for hydrogen adsorption. For guiding the structure–performance relationship, FAM yield is also plotted as a function of Ru–N coordination structure.

structure, a similar increasing trend can be found as that of hydrogen adsorption heat (Fig. 6a), which suggests that moderate hydrogenation capability, as well as strong transimination activity of the catalyst, is highly desirable for the selective synthesis of primary amines. Kinetic isotope effect (KIE) experiment by using $D_2$ to replace $H_2$ in the reductive amination reveals KIE value of 1.1, 1.7, and 2.1 for the $Ru_1$/NC-800, $Ru_1$/NC-900, and $Ru_1$/NC-900-800$NH_3$ catalysts (Supplementary Table 5), respectively, proving that hydrogen activation is an important factor determining the product selectivity although it is not involved in the rate-determining step.

To get more insight into the effect of the relative strength between hydrogenation and transimination capabilities on the product selectivity, we modulate the hydrogenation/transimination activities by varying the pressure of $H_2$/$NH_3$. As shown in Supplementary Fig. 13a, when the $H_2$ pressure decreases from 2 to 1 MPa, the FAM yield is correspondingly reduced to 41%, while the yield of oligomer increases to 18% due to the reduced hydrogenation capability. By contrast, when the $NH_3$ pressure falls from 0.5 to 0.2 MPa (Supplementary Fig. 13b), 9% secondary amine is yielded at the expense of primary amine (yield: 68%). Lower pressure of $NH_3$ results in weakened transimination activity yet enhanced hydrogenation capability due to competitive adsorption between $NH_3$ and $H_2$, which leads to the hydrogenation of Schiff base to give secondary amine.

Since the reaction direction (selectivity) is steered by the relative adsorption strength between $NH_3$ and $H_2$, and this factor can be modulated by the Ru–$N_x$ coordination sphere, it is important to elucidate how $H_2$ is activated by the Ru–$N_x$. Based on our previous studies on the Ni–$N_x$[36] and Pt–$O_x$[33] SACs, we suppose the central Ru and the neighboring N atoms form Frustrated Lewis Pairs (FLPs)[55,56], which is able to dissociate $H_2$ following a heterolysis manner. According to this FLP mechanism, pyrrole and potassium thiocyanate (KSCN) were selected as poisons of Lewis basic and Lewis acid sites, respectively. As shown in Supplementary Fig. 14, upon the addition of pyrrole or KSCN, the yield of primary amine decreases significantly from 97 to 21% or 1%, respectively, which provides strong evidence that FLP is required for hydrogen dissociation. In addition, $CO_2$ temperature-programmed desorption ($CO_2$-TPD) experiments (Supplementary Fig. 15) and $CO_2$ adsorption measurements (Supplementary Fig. 16) show that both of the adsorption strength and uptake of $CO_2$ are greatly enhanced after $NH_3$ treatment of $Ru_1$/NC-900, suggesting the higher basicity of the $Ru_1$/NC-900–800$NH_3$ catalyst, which is in line with that reported previously[57,58]. Also, this is in agreement with the XPS result that reveals the increased electron density on the N atoms

coordinating to Ru. The increased basicity in FLPs facilitates the heterolytic cleavage of $H_2$ according to our previous DFT calculations[33] and thus contributes to the high hydrogenation activity.

In summary, we have fabricated highly active, selective and robust Ru SACs supported on N-doped carbons using a simple pyrolysis approach. By raising the pyrolysis temperature from 700 to 900 °C, the Ru–N coordination number decreases from $RuN_5$, $RuN_4$ to $RuN_3$, meanwhile, the electron density of the Ru single atoms increases in the order of $RuN_5 < RuN_4 < RuN_3$. By further treatment with $NH_3$ at 800 °C, the electron density on the $RuN_3$ site is further increased. As a consequence of the fine-tuning in both the coordination structure and the electron density of the Ru single atoms, the catalytic performance is finely modulated in the reductive amination of aldehydes/ketones. Among them, the $Ru_1$/NC-900–800$NH_3$ SAC affords the highest activity and selectivity to primary amines thanks to its moderate capability for $H_2$ activation in excess of $NH_3$. Moreover, the catalyst exhibits excellent stability in reuse tests and tolerance to a wide spectrum of biomass-derived aldehydes/ketones substrates. In particular, it shows superior robustness against being poisoned by CO, sulfur, or harsh hydrogen treatment which, by contrast, drastically deactivate the control Ru/AC nanocatalyst. The structure–performance relationship established in this work will open an avenue toward the rational design of highly selective and robust SACs for other demanding transformations.

## Methods

**Sample preparation**. Typically, a mixture of dicyandiamide ($C_2H_4N_4$) (12 g), L-cysteine ($C_3H_7NO_2S$) (3 g) and Ru(III) acetylacetonate (Ru(acac)$_3$) (40 mg) were added to a ball mill pot, and were ground at a speed of 400 rpm for 2.0 h for twice at room temperature. The obtained fine powder was then subjected to temperature-programmed pyrolysis in a tubular furnace under $N_2$ atmosphere with a flow rate of 60 ml min$^{-1}$. First stage: from 25 to 600 °C at a ramping rate of 3 °C min$^{-1}$, maintained at 600 °C for 2 h; second stage: from 600 to 900 °C at a ramping rate of 2 °C min$^{-1}$, maintained at 900 °C for 1 h. The as-made sample was denoted as $Ru_1$/NC-900. Other $Ru_1$/NC samples were synthesized by the same procedure except for different maximum temperatures. $Ru_1$/NC-900-800$NH_3$ was prepared by subjecting $Ru_1$/NC-900 to pyrolysis at 800 °C for 30 min (ramping rate: 5 °C min$^{-1}$) under $NH_3$/He mixture atmosphere (30 ml min$^{-1}$ He and 20 ml min$^{-1}$ $NH_3$).

More sample preparation details are described in the Supplementary Methods section.

**Sample characterization**. The actual Ru loadings were determined by inductively coupled plasma spectroscopy (ICP-OES) on an IRIS Intrepid II XSP instrument (Thermo Electron Corporation). X-ray diffraction (XRD) patterns were recorded on a PANalytical X' pert diffractometer with a Cu–$K_\alpha$ radiation source (40 kV and 40 mA). A continuous mode was used to record data in the 2$\theta$ range from 10° to 80°. $N_2$ adsorption-desorption experiments were conducted on a Micromeritics

ASAP-2010 physical adsorption apparatus. The specific surface area was calculated using a Brunauer–Emmett–Teller (BET) method.

Scanning transmission electron microscopy (STEM) and energy-dispersive X-ray spectroscopy (EDS) experiments were performed on a JEOL JEM-2100F microscope operated at 200 kV, equipped with an Oxford Instruments ISIS/INCA energy-dispersive X-ray spectroscopy (EDS) system with an Oxford Pentafet Ultrathin Window (UTW) Detector. The aberration-corrected high-angle annual dark-filed scanning transmission electron microscopy (AC-HAADF-STEM) analysis was performed on a JEOL JEM-ARM200F equipped with a CEOS probe corrector, with a guaranteed resolution of 0.08 nm. The Electron Energy Loss Spectroscopy (EELS) analysis of $Ru_1/NC$-900-800$NH_3$ was performed on a Field Emission HF5000 Microscope (200 kV accelerating voltage; aberration corrector, 0.078-nm spatial resolution). Before microscopy examination, the sample was ultrasonically dispersed in ethanol for 15–20 min, and then a drop of the suspension was dropped on a copper TEM grid coated with a thin holey carbon film.

X-ray photoelectron spectroscopy (XPS) spectra were obtained on a Thermo ESCALAB 250 X-ray photoelectron spectrometer equipped with Al Kα excitation source and with C as internal standard (C 1 s = 284.6 eV).

The X-ray absorption spectra (XAS) including X-ray absorption near-edge structure (XANES) and extended X-ray absorption fine structure (EXAFS) at Ru $K$-edge of the samples were measured at the beamline 14 W of Shanghai Synchrotron Radiation Facility (SSRF) in China. The output beam was selected by Si(311) monochromator, and the energy was calibrated by Ru foil. The data were collected at room temperature under transmission mode. Athena software package was employed to process the XAS data.

Microcalorimetric measurement was performed by a BT2.15 heat-flux calorimeter, which was connected to a gas handling and a volumetric system employing MKS Baratron Capacitance Manometers for precision pressure measurement. The ultimate dynamic vacuum of the microcalorimetric system was $10^{-7}$ Torr by calculation. First, the fresh sample was treated in a special tube in $H_2$ at 100 °C for 1 h and then high pure He at 200 °C for 1 h to eliminate the adsorption. Then, the tube was transferred into the high vacuum system and stabilized for (6–8 h). After thermal equilibrium was reached, the $H_2$-microcalorimetric data were collected by sequentially introducing small doses ($10^{-6}$ mol) of $H_2$ ($CO_2$ or $NH_3$) into the system until it became saturated (5–6 Torr). Simultaneously, the differential heat versus adsorbate coverage plots and adsorption isothermals can be obtained after a typical microcalorimetric experiment.

More characterization performance details are described in the Supplementary Methods section.

**Catalytic test**. The reductive amination reaction was conducted in a 50-mL stainless-steel autoclave (Parr Instrument Company, America). Typically, the substrate (2 mmol), $Ru_1/NC$ catalyst (molar ratio of Ru:substrate is 1:400), dodecane (internal standard) and 3 mL methanol (solvent) were added to a teflon lining. Then the autoclave was sealed and purged by $N_2$ for three times, followed by charging with 0.5 MPa $NH_3$ and 2 MPa $H_2$. The reaction mixture was stirred at 100 °C for 10 h. The products were identified using a Varian G450/320 GC/MS system and were quantitatively analyzed using an Agilent 7890 A GC system equipped with a HP-5 capillary column and an FID detector.

The furfural conversion (1) and furfural amine selectivity (2) for furfural reductive amination reaction were calculated as follows:

$$\text{Furfural conversion}(\%) = \frac{n(\text{fufural})_{\text{fed}} - n(\text{furfural})_{\text{consumed}}}{n(\text{fufural})_{\text{fed}}} \times 100\% \quad (1)$$

$$\text{Furfural amine selectivity}(\%) = \frac{n(\text{furfural amine})_{\text{produced}}}{n(\text{fufural})_{\text{fed}}} \times 100\% \quad (2)$$

## Data availability

All data are available within the article, and its supplementary information file is available from the authors upon request.

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

## Acknowledgements

This work was supported by the National Key R&D Program of China (2018YFB1501602), the National Natural Science Foundation of China (21690080, 21690084, 21673228, 21721004, and 21878289), the Strategic Priority Research Program of the Chinese Academy of Sciences (XDB17020100), and the Dalian National Laboratory for Clean Energy (DNL) Cooperation Fund, the CAS (DNL 180303). We are grateful to Dr. Wentao Wang and Mr. Yiqi Ren for their kind help in products isolation and NMR discussion.

## Author contributions

H.Q. and J.Y. prepared the catalysts. H.Q. performed reaction tests and most of the characterizations. F.L. performed a part of characterizations and helped to revise the manuscript. J.Y.Y. and R.H. helped to do XPS and NMR data analysis. X.L. helped to do XAS measurement and data fitting. L.L. helped to do microcalorimetric measurements. Y.S. and Y.L. helped to do HAADF-STEM and EELS characterization. A.W. and T.Z. conceived the idea and directed the project. H.Q., L.Z., and A.W. co-wrote the manuscript. All the authors discussed the results and commented on the manuscript.

## Competing interests

The authors declare no competing interests.
