## [Peer Review File · Nature Communications]

REVIEWER COMMENTS

Reviewer #1 (Remarks to the Author):

The authors have reported Ru single atom/C to exhibit high catalytic performance for the reductive amination. However, this manuscript is insufficient to demonstrate the advantage of Ru single atom catalysts at present for the followings.

1. I am wondering whether the catalytic activities of other Ru catalytic systems are compared with that of the material presented in this manuscript in Table S3; the reader cannot conclude that Ru single atom/C is superior to other Ru catalytic systems only from Table S3.

Table S3 comes into its own when the formation rate at the early stage of the reaction for each catalyst is measured under the same reaction conditions (reactant concentration, temperature, H₂ pressure and so on). Such information is not shown in the main text and supporting information.

2. I cannot think that Ru single atom/C is of greater advantage than other Ru catalytic systems even if present Table S3 exactly compare the activities of Ru catalysts.

In Ru nanoparticles and Ru-deposited support catalysts, Ru nanoparticles of several nm act as active sites for the reaction. Most Ru atoms in the Ru particles, bulk Ru atoms, cannot participate in the reaction, nevertheless the formation rates (gFAM gRu⁻¹ h⁻¹) reach ca. 120 in Table S3. On the other hand, the formation rate for Ru single atom/C is 171 although most Ru atoms can participate in the reaction in the case of this catalytic system. This difference in formation rate between the former and latter is too small to demonstrate the advantage of single atom catalyst.

Reviewer #2 (Remarks to the Author):

Zhang and co-workers reported the preparation of Ru-based single atom catalysts and their potential application for the reductive amination of carbonyl compounds with ammonia to produce primary amines. For the modern-state-of-the-art catalysts, the development of single atom-based catalysts is of prime importance, which provides opportunities to combine both homogeneous and heterogeneous catalysis aspects. Thus, in recent years single atom-based catalysts emerged as promising catalysts for the advancement of sustainable chemical synthesis. In this respect, authors have developed promising Ru-based single atoms, which constitute promising catalysts for the industrially relevant reductive amination reaction to synthesize various primary amines. Noteworthy, the prepared Ru-SACs were systematically characterized by modern techniques. The catalyst development and characterization parts in this manuscript are of high potential and very interesting. However, the general applicability of these Ru-SACs for the reductive amination part must be strengthened to make this paper more interesting and potential both in single-atom catalysis and organic synthesis research areas. In this respect, it is suggested to perform reductive amination of more challenging, functionalized and structurally diverse carbonyl compounds to prepare corresponding primary amines in selective manner. Yields of the products were reported by GC analysis; but isolated yields for selected products should be

given including NMR data and spectra. Interesting to note that authors have performed catalyst poisoning/deactivation studies. However, catalyst stability experiments are missing. Hence, its suggested to demonstrate catalyst stability by performing catalyst recycling for around 50% conversions like that made for full conversion. The chemical structures in Table 2, 3 and in other places both in the manuscript and supporting information are not properly drawn. Chemical structures should be corrected with proper orientation and bond length. In table 2 it was given Ru: furfuraldehyde = 1: 400 or 1:200; it should be in general substrate instead of furfuraldehyde. What does it means, Ru: furfuraldehyde = 1: 400 or 1:200? is this ration related to weight of substrate or catalyst amount (or Ru content) or mmol of substrate: mmol of catalyst (Ru-content)? To be clear, it would be better to give the weight of catalyst used for the reaction and the mol% of Ru. The imidazole-based product 3a, was observed in case of furaldehyde or commonly this type of product forms in other aldehydes too? It would be better to check in few aldehydes also. In the reductive amination of furaldehyde (Table1), formation of desired product and only side product 3a is mentioned. But there should be the formation of secondary imine, secondary amine and corresponding alcohol. This need to be checked properly and these products should be included in the reaction scheme and the yields in the table. This work is quite interesting, but authors needs to demonstrate the applicability of their catalyst in more general sense by performing additional/broad substrate scope, and other experiments suggested above. Hence this manuscript requires suitable revision.

Reviewer #3 (Remarks to the Author):

The authors report on a Ru single metal atom catalyst for the reductive amination. With regard to the reductive amination, the synthesis of primary amines, a challenging but highly interesting reaction was investigated. A rather low metal loading could be used but reaction conditions such as pressure of H₂ and temperature are relatively high. The addressed scope is interesting but the demonstrated tolerance of functional groups seems rather limited, may be at this stage. Single metal atom catalysis is meanwhile a rather mature field of research and catalyst development should lead to catalysts with an outstanding performance. Note, the catalyst synthesis strategy is not new (reference 36). Unfortunately, the authors missed to describe the state of the art comprehensively. The most comprehensive review (Chem. Rev. 2020) is not cited as recent development in the field of the synthesis of primary amines via reductive amination.

At this stage, I can only weakly support publication and hope to see a revised version with a comprehensive discussion of the state of the art and a more impressive tolerance of functional groups.

Minor:

Scheme 1 is overdone!

Many figures contain very small numbers and/or letters and need to get redesigned.

Rhett Kempe

Responses to the referees

We are very grateful to the three reviewers for their careful reading of the manuscript and raising critical comments and constructive suggestions which are definitely helpful to the improvement of the paper quality. The point-by-point response to the reviewers are shown below.

Reviewer # 1:

The authors have reported Ru single atom/C to exhibit high catalytic performance for the reductive amination. However, this manuscript is insufficient to demonstrate the advantage of Ru single atom catalysts at present for the followings.

1. I am wondering whether the catalytic activities of other Ru catalytic systems are compared with that of the material presented in this manuscript in Table S3; the reader cannot conclude that Ru single atom/C is superior to other Ru catalytic systems only from Table S3. Table S3 comes into its own when the formation rate at the early stage of the reaction for each catalyst is measured under the same reaction conditions (reactant concentration, temperature, H₂ pressure and so on). Such information is not shown in the main text and supporting information.

Answer: We appreciate the reviewer's critical comment on the activity comparison of Ru₁/NC-900-800NH₃ catalyst in the present work with other catalysts reported in literature. In the original submission we only compared our catalyst with Ru catalysts reported in literature for the reductive amination of furfural. Now, the comparison is extended to all the Ru catalysts we have searched in the literature for the reductive amination of other aldehydes/ketones (see the revised **Table S3** below). You can see that our Ru₁/NC-900-800NH₃ single-atom catalyst is several times to an order of magnitude more active than a variety of reported Ru catalysts for the reductive amination of aldehydes. Moreover, in order to make a strict comparison, we have prepared Ru/Nb₂O₅ and Ru/HZSM-5 catalysts which had been reported to be highly active for the reductive amination of furfural (*J. Am. Chem. Soc.* 2017, **139**,

11493-11499; *Mol. Cat.* 2020, **482**, 110755) and tested their catalytic activities under identical conditions to our Ru₁/NC-900-800NH₃ catalyst. The results, together with the reaction conditions (temperature, pressure and time) are summarized in the revised **Table S3** below. As you see, for the reductive amination of furfural, both the Ru/Nb₂O₅ and Ru/HZSM-5 nanocatalysts gave lower yields of furfurylamine (FAM) than our Ru₁/NC-900-800NH₃ single-atom catalyst even in extended reaction time (entry 1-3 in Table S3). The calculation of the production rate shows that it is 115 and 89.6 g_{FAM}·g_{Ru}⁻¹·h⁻¹ for the Ru/Nb₂O₅ and Ru/HZSM-5, respectively, both of them are much lower than that (170.7 g_{FAM}·g_{Ru}⁻¹·h⁻¹) over our Ru₁/NC-900-800NH₃ single-atom catalyst. Based on the above strict comparison under identical reaction conditions, we can conclude that the Ru₁/NC-900-800NH₃ single-atom catalyst is indeed more active than those best-performance nanocatalysts reported in literature.

Table S3. The superior performance of Ru₁/NC-900-800NH₃ as compared to the reported Ru and Rh nanocatalysts.^a

Entry	Catalyst	Molar ratio Ru/Substrate	P _{NH₃} (MPa)	P _{H₂} (MPa)	Time (h)	Tem. (°C)	Amine yield (%)	Production Rate (g _{FAM} ·g _{Ru} ⁻¹ ·h ⁻¹)	Ref.
1	Ru ₁ /NC-900-800NH ₃	1:400	0.5	2	4	100	97	170.7	This work
2 ^b	Ru/Nb ₂ O ₅	1:400	0.5	2	6	100	91	115	This work
3 ^c	Ru/HZSM-5	1:400	0.5	2	7.5	100	68	89.6	This work
4 ^d	Ru ₁ /NC/Nb ₂ O ₅	1:400	0.5	2	3	100	97	227.6	This work
5	Ru/Nb ₂ O ₅	1:250	0.1	4	2	90	99	124.2	1
6	RuCl ₂ (PPh ₃) ₃	1:50	0.5-0.7	4	24	130	85	2	2
7	Ru nanoparticles	1:250	8 mmol	2	1.5	90	99	125.2	3
8	Ru/TiP	1:500	0.3	1.7	24	30	91	18.2	4
9	Ru/HZSM-5	1:27	7 mmol	3	0.25	100	76	103.5	5
10	Ru/Nb ₂ O ₅ -L	1:1000	10 mmol	2	8	90	60	72.8	6
11	Ru/Nb ₂ O ₅ ·nH ₂ O-300	1:250	8 mmol	4	4	70	89	54.6	7
12	Ru/SiO ₂	1:670	1mL NH ₄ OH	7.5	4	130	90	145.5	8
13	Ru/γ-Al ₂ O ₃	1:266	0.4	3	2	80	75	96	9
14 ^e	Ru/ZrO ₂	1:40	29 mmol	2	12	85	94	1.9	10

15 ^f	Ru/C	1:114	17.6mmol	1.5	14	100	95	13.8	11
16 ^g	Ru/C	1:100	0.4	0.9	2	25	87.8	42.2	12
17	Rh/Al ₂ O ₃	1:2000	70mmol	2	2	80	91.5	45.8	13

^a Unless otherwise noted, the substrate was furfural. ^b The catalyst was prepared based on the procedure reported in *J. Am. Chem. Soc.* 2017, **139**, 11493-11499; ^c The catalyst was prepared based on the procedure reported in *Mol. Cat.* 2020, **482**, 110755; ^d The catalyst was prepared with the similar procedure to that for Ru₁/NC-900-800NH₃ catalyst except for the employment of Nb₂O₅ support. ^e Glycolaldehyde as substrate; ^f (E)-1-(furan-2-yl)-5-methylhex-1-en-3-one as substrate; ^g Cyclohexanone as substrate.

In the revised manuscript, we have updated **Table 2** as below, and made the corresponding revision of the related text as follows:

“We also prepared other nano-catalysts including Ru/AC, Ru/Nb₂O₅ and Ru/HZSM-5 which were reported to be highly active for the reductive amination of furfural.^{19, 45-46} Under identical reaction conditions, the Ru/AC nano-catalyst gives an enhanced FAM yield (82%, Table 2, entry 6) than Ru₁/NC-1000, which can be due to the smaller and more exposed Ru particles on the AC support (Figure S7), yet lower than that over the Ru₁/NC-900-800NH₃. Similarly, both the Ru/Nb₂O₅ and Ru/HZSM-5 nanocatalysts deliver inferior yield of FAM to the Ru₁/NC-900-800NH₃ SAC even with extended reaction time (Table 2, entry 7, 8). To further demonstrate the superiority of the Ru₁/NC SAC, we synthesized the Ru₁/NC SAC on the Nb₂O₅ support given that the electronic interaction between Ru and acidic and reducible Nb₂O₅ support would facilitate the reductive amination reaction.^{19, 25} To our delight, the as-prepared Ru₁/NC/Nb₂O₅ SAC affords 97% yield in a shorter time of 3 h (Table 2, entry 9), resulting in a production rate of 227.6 g_{FAM}·g_{Ru}⁻¹·h⁻¹, almost two-fold higher than that on the Ru/Nb₂O₅ nanocatalyst. These strictly control experiments, together with the comparison with more Ru catalysts reported in literature (Table S3), unequivocally demonstrate the remarkable advantage of Ru₁/NC-900-800NH₃ SAC in maximizing the atom efficiency of Ru.”

Table 2. Catalytic performances of Ru₁/NC-T SACs as well as reference Ru nanocatalysts for the reductive amination of furfural. ^a

Entry	Catalysts	Yield (%) ^d			Production Rate (g _{FAM} ·g _{Ru} ⁻¹ ·h ⁻¹)
		2a	3a	Others	
1	Ru ₁ /NC-700	n.d.	33	67	-
2	Ru ₁ /NC-800	43	9	48	16.7
3	Ru ₁ /NC-900	84	9	7	81.5
4	Ru ₁ /NC-900-800NH ₃	97 (94 [°])	3	n.d.	170.7
5	RuNP-1000	53	12	33	20.6
6	Ru/AC	82	12	6	79.5
7	Ru/Nb ₂ O ₅	91	3	6	115.0
8	Ru/HZSM-5	68	12	20	89.6
9 ^b	Ru ₁ /NC/Nb ₂ O ₅	97	n.d.	3	227.6

^a Reaction conditions: 2 mmol furfural, catalysts amounts were varied to maintain the molar ratio of Ru: furfural = 1: 400, 3 g methanol, 0.5 MPa NH₃, 2 MPa H₂, 100 °C, 10 h, dodecane as an internal standard. The conversion of FAL in all experiments were >99%; ^b 3 h; ^c Isolated yield in parentheses; ^d Others are oligomers; n.d.: not detected.

2. I cannot think that Ru single atom/C is of greater advantage than other Ru catalytic systems even if present Table S3 exactly compare the activities of Ru catalysts. In Ru nanoparticles and Ru-deposited support catalysts, Ru nanoparticles of several nm act as active sites for the reaction. Most Ru atoms in the Ru particles, bulk Ru atoms, cannot participate in the reaction, nevertheless the formation rates ($\text{g}_{\text{FAM}} \cdot \text{g}_{\text{Ru}}^{-1} \cdot \text{h}^{-1}$) reach ca. 120 in Table S3. On the other hand, the formation rate for Ru single atom/C is 171 although most Ru atoms can participate in the reaction in the case of this catalytic system. This difference in formation rate between the former and latter is too small to demonstrate the advantage of single atom catalyst.

Answer: I respectfully don't agree on this point. Compared to the NPs counterpart, the advantage of single-atom catalyst has been demonstrated in two aspects, one is the superior metal utilization efficiency, and the other is the ultra-robustness against being poisoned by CO, sulfur, as well as harsh hydrogen treatment.

For the superior metal atom utilization efficiency, we have demonstrated that the production rate of FAM over the Ru₁/NC-900-800NH₃ catalyst is 170.7 $\text{g}_{\text{FAM}} \cdot \text{g}_{\text{Ru}}^{-1} \cdot \text{h}^{-1}$ whereas it is only 115.0 $\text{g}_{\text{FAM}} \cdot \text{g}_{\text{Ru}}^{-1} \cdot \text{h}^{-1}$ over the Ru/Nb₂O₅ nanocatalyst under the identical reaction conditions. Obviously, our single-atom catalyst has a 1.5-fold increase based on Ru utilization efficiency compared to Ru/Nb₂O₅ nanocatalyst, and this should be a significant enhancement and will lead to a nonignorable reduction in the catalyst cost given that Ru is expensive (~15 USD/g). Moreover, it should be noted that the catalytic activity of the reported Ru/Nb₂O₅ nanocatalyst arose from the synergistic effect between Ru and the reducible Nb₂O₅ support (see *J. Am. Chem. Soc.* 2017, **139**, 11493-11499). Therefore, it can be reasonably speculated that a higher activity might be achieved if the Ru₁/NC single-atom catalyst is supported on the Nb₂O₅ carrier. In order to prove this speculation, we synthesized our Ru₁/NC single-atom catalyst on the Nb₂O₅ support by using the same procedure as that for Ru₁/NC-900-800NH₃. The activity evaluation test showed that the as-prepared Ru₁/NC/Nb₂O₅ afforded production rate of 227.6 $\text{g}_{\text{FAM}} \cdot \text{g}_{\text{Ru}}^{-1} \cdot \text{h}^{-1}$ (see the above Table 2, entry 9), a nearly 2-fold increase than the Ru/Nb₂O₅ nanocatalyst. This control

experiment unambiguously demonstrates that the single-atom catalyst is superior to the NPs counterpart in terms of metal utilization efficiency. I hope this big difference can convince you.

For the second advantage of single-atom catalyst, that is, the ultra-robustness, we have provided compelling evidence in the original submission that our Ru₁/NC-900-800NH₃ single-atom catalyst exhibits outstanding resistance against being poisoned by CO, sulfur, as well as harsh hydrogen treatment, far superior to the Ru/AC nanocatalyst. In order to further demonstrate this advantage, we now compare it with the reported best-performance catalyst Ru/Nb₂O₅. As shown in the figure below (**Figure S9**), similar to the Ru/AC nanocatalyst, the Ru/Nb₂O₅ nanocatalyst is also subject to being poisoned by CO and sulfur leading to significant decrease of FAM yield. On the contrary, our Ru₁/NC-900-800NH₃ single-atom catalyst is only slightly affected by these typical poisons. For practical applications, the catalyst robustness is by no means less important than activity, especially for large-scale production.

With these newly added data, we hope that we can convince you that our Ru₁/NC single-atom catalysts are significantly advantageous over the state-of-the-art nanocatalysts reported thus far. In the revised manuscript, we have updated **Table 2**, **Table S3**, **Figure S9** and made the corresponding revision of the related text.

Figure S9. Poisoning experiments of Ru₁/NC-900-800NH₃ and Ru/Nb₂O₅ by CO and sulfur.

Reaction condition: 2 mmol furfural, 22 mg Ru₁/NC-900-800NH₃ or 51 mg Ru/Nb₂O₅ (0.25mol% Ru, molar ratio of Ru: furfural = 1: 400), 3 g methanol, 0.5 MPa NH₃, 2 MPa H₂, 100 °C, 10 h. For CO poisoning experiment, 2 MPa 1vol% CO in H₂ was used in place of pure H₂; for sulfur poisoning experiment, 500 ppm thiophene was added to the reaction mixture.

Reviewer #2

1. Zhang and co-workers reported the preparation of Ru-based single atom catalysts and their potential application for the reductive amination of carbonyl compounds with ammonia to produce primary amines. For the modern-state-of-the-art catalysts, the development of single atom-based catalysts is of prime importance, which provides opportunities to combine both homogeneous and heterogeneous catalysis aspects. Thus, in recent years single atom-based catalysts emerged as promising catalysts for the advancement of sustainable chemical synthesis. In this respect, authors have developed promising Ru-based single atoms, which constitute promising catalysts for the industrially relevant reductive amination reaction to synthesize various primary amines. Noteworthy, the prepared Ru-SACs were systematically characterized by modern techniques. The catalyst development and characterization parts in this manuscript are of high potential and very interesting.

Answer: We appreciate very much for the reviewer's positive comment.

2. However, the general applicability of these Ru-SACs for the reductive amination part must be strengthened to make this paper more interesting and potential both in single-atom catalysis and organic synthesis research areas. In this respect, it is suggested to perform reductive amination of more challenging, functionalized and structurally diverse carbonyl compounds to prepare corresponding primary amines in selective manner.

Answer: Thanks for this good suggestion. Based on this suggestion, we have extended the original substrates scope to more challenging and structurally diverse carbonyl compounds (15 newly added substrates), and these new results are added in the updated **Table 3** in the revised manuscript (see highlighted entries). As you see from the **Table 3** below, for all these aldehydes/ketones substrates with diverse

structures and functional groups, our Ru₁/NC-900-800NH₃ single-atom catalyst worked well and afforded moderate to good yields, demonstrating the general applicability of the Ru₁/NC single-atom catalyst.

Table 3. Production of primary amines from various aldehydes and ketones over Ru₁/NC-900-800NH₃ catalyst. ^a

Entry	Substrate	Product	Yield (%)
1			98
2			93
3 ^b			63
4		97	
5			90
6			99
7			99
8			93
9			99
10 ^c			91 (87 ^h)
11 ^c			81
12			95 (90 ^h)
13			91 (88 ^h)
14			91
15 ^d			85

16			92 (90 ^h)
17			93
18			95
19 ^d			67
20 ^e			81
21 ^e			90
22			93
23			67/28
24 ^f			95
25 ^f			96 (91 ^h)
26 ^f			91
27 ^f			81
28 ^f			94(90 ^h)
29 ^f			89 (85 ^h)
30 ^f			73
31 ^f			71

32 ^f			65
33			96
34			65
35			91
36 ^g			61

^a Reaction condition: 2 mmol substrate, 22 mg Ru₁/NC-900-800NH₃ catalyst, 0.25 mol% Ru, molar ratio of Ru: substrate = 1: 400, 3 g methanol, 0.5 MPa NH₃, 2 MPa H₂, 100 °C, 10 h, dodecane as an internal standard. ^b Adding 6 mmol butylamine; ^c 80 °C; ^d 44 mg Ru₁/NC-900-800NH₃ catalyst, 0.5 mol% Ru, molar ratio of Ru: substrate = 1: 200, 120 °C; ^e 3 g 25 wt% aqueous ammonia as solvent, 2 MPa H₂, 80 °C, 10 h; ^f 1 MPa NH₃; ^g 5 g p-xylene as solvent, 0.8 MPa NH₃, 0.2 MPa H₂, 180 °C, 20 h; ^h Isolated yields in parentheses for some selected amine products and NMR and HR-MS spectra are shown in Figures S18-24.

3. Yields of the products were reported by GC analysis; but isolated yields for selected products should be given including NMR data and spectra.

Answer: The isolated yields of selected products as well as their NMR data and spectra have been provided in the revised manuscript (Table 2, 3, figures in brackets) and Supporting Information (Figs. S17-S24).

¹H NMR (400 MHz, DMSO-*d*₆) δ 8.74 (s, 3H), 7.73 (dd, *J* = 1.9, 0.8 Hz, 1H), 6.59 (dd, *J* = 3.3, 0.8 Hz, 1H), 6.50 (dd, *J* = 3.3, 1.8 Hz, 1H), 4.05 (q, *J* = 5.8 Hz, 2H). ¹³C NMR (101 MHz, DMSO) δ 148.05,

143.95, 111.36, 110.76, 35.35. FT-ICR-MS (m/z): Calcd for $C_5H_8NO^+$ $[M+H]^+$ 98.0600; found 98.0601.

1H NMR (400 MHz, $DMSO-d_6$) δ 8.08 (s, 3H), 7.38 (m, $J = 9.0, 5.7, 2.9$ Hz, 2H), 7.11 – 7.00 (m, 2H), 3.95 (q, $J = 5.9$ Hz, 2H). ^{13}C NMR (101 MHz, $DMSO$) δ 163.78, 161.34, 131.65, 131.57, 129.31, 129.28, 116.10, 115.89, 42.37. FT-ICR-MS (m/z): Calcd for $C_7H_9FN^+$ $[M+H]^+$ 126.07135; found 126.07140.

1H NMR (400 MHz, $DMSO-d_6$) δ 8.49 (s, 3H), 7.04 (d, $J = 2.1$ Hz, 1H), 6.93 (dd, $J = 8.3, 2.1$ Hz, 1H), 6.83 (d, $J = 8.3$ Hz, 1H), 4.20 (s, 4H), 3.85 (q, $J = 5.8$ Hz, 2H). ^{13}C NMR (101 MHz, $DMSO$) δ 143.91, 143.55, 127.29, 122.55, 118.37, 117.47, 64.55, 64.51, 42.08. FT-ICR-MS (m/z): Calcd for $C_9H_{12}NO_2^+$ $[M+H]^+$ 166.08626; found 166.08637.

1H NMR (700 MHz, $DMSO-d_6$) δ 10.41 (s, 1H), 8.58 (d, $J = 6.5$ Hz, 3H), 7.62 (d, $J = 8.4$ Hz, 2H), 7.40 (d, $J = 8.4$ Hz, 2H), 3.88 (d, $J = 5.9$ Hz, 2H), 2.03 (s, 3H). ^{13}C NMR (176 MHz, $DMSO$) δ 169.06, 139.94, 130.79, 129.94, 128.70, 124.00, 119.27, 42.21, 24.44. FT-ICR-MS (m/z): Calcd for $C_9H_{13}N_2O^+$ $[M+H]^+$ 165.10224; found 165.1024.

¹H NMR (400 MHz, DMSO-*d*₆) δ 8.56 (s, 3H), 7.45 (d, *J* = 8.4 Hz, 4H), 7.39 (t, *J* = 7.2 Hz, 2H), 7.35 – 7.30 (m, 1H), 7.03 (dd, *J* = 8.7, 2.8 Hz, 2H), 5.13 (d, *J* = 3.0 Hz, 2H), 3.92 (d, *J* = 5.8 Hz, 2H). ¹³C NMR (101 MHz, DMSO) δ 158.77, 137.43, 131.05, 130.88, 128.90, 128.29, 128.13, 128.08, 126.72, 115.36, 115.26, 69.63, 42.05. FT-ICR-MS (*m/z*): Calcd for C₁₄H₁₆NO⁺ [M+H]⁺ 214.12264; found 214.12333.

¹H NMR (400 MHz, DMSO-*d*₆) δ 9.32 (s, 3H), 7.72 – 7.52 (m, 4H), 7.41 (t, *J* = 7.5 Hz, 4H), 7.37 – 7.27 (m, 2H), 5.63 (s, 1H). ¹³C NMR (101 MHz, DMSO) δ 138.91, 129.16, 128.67, 127.86, 57.58. FT-ICR-MS (*m/z*): Calcd for C₁₃H₁₄N⁺ [M+H]⁺ 184.11208; found 184.11220.

¹H NMR (400 MHz, DMSO-*d*₆) δ 8.66 (s, 3H), 7.57 – 7.37 (m, 2H), 7.02 – 6.82 (m, 2H), 3.73 (s, 3H), 1.50 (d, *J* = 6.8 Hz, 3H). ¹³C NMR (101 MHz, DMSO) δ 159.59, 131.78, 128.78, 114.36, 55.67, 50.02, 21.20. FT-ICR-MS (*m/z*): Calcd for C₉H₁₄NO⁺ [M+H]⁺ 152.10699; found 152.10711.

^1H NMR (400 MHz, $\text{DMSO-}d_6$) δ 8.81 (s, 3H), 7.66 – 7.57 (m, 2H), 7.53 – 7.44 (m, 2H), 4.41 (p, J = 6.1 Hz, 1H), 1.53 (d, J = 6.8 Hz, 3H). ^{13}C NMR (101 MHz, DMSO) δ 138.89, 133.36, 129.44, 129.01, 49.83, 21.12. FT-ICR-MS (m/z): Calcd for $\text{C}_8\text{H}_{11}\text{ClN}^+$ $[\text{M}+\text{H}]^+$ 156.05745; found 156.05768.

Figure S17-S24. ^1H NMR, ^{13}C NMR and FT-ICR-MS spectra of some representative amine products.

4. Interesting to note that authors have performed catalyst poisoning/deactivation studies. However, catalyst stability experiments are missing. Hence, it is suggested to demonstrate catalyst stability by performing catalyst recycling for around 50% conversions like that made for full conversion.

Answer: Based on this suggestion, we performed additional catalyst recycling test at around 60% yield of FAM. As you see in the updated **Figure 4**, the $\text{Ru}_1/\text{NC-900-800NH}_3$ catalyst showed excellent stability, without appreciable decrease in the FAM yield. In the revised manuscript, the recycling results at both conditions, i.e., 97% and 60% yield of FAM, have been presented in **Figure 4**, as shown below.

Figure 4. Reusability (a) and resistance against CO, sulfur and H_2 treatment (b) of $\text{Ru}_1/\text{NC-900-800NH}_3$ and Ru/AC . Reaction conditions: 2 mmol furfural, 22 mg $\text{Ru}_1/\text{NC-900-800NH}_3$ or Ru/AC catalysts (0.25mol% Ru, molar ratio of Ru: furfural = 1: 400), 3 g methanol, 0.5 MPa NH_3 , 2 MPa H_2 , 100 $^\circ\text{C}$. For reusability tests, data were taken at 1 h (left) and 10 h (right), respectively. For CO poisoning experiment, 2 MPa 1vol% CO in H_2 was used in place of pure H_2 ; for sulfur poisoning experiment, 500 ppm thiophene was added to the reaction mixture; for H_2 treatment test, the catalysts were pre-reduced at 600 $^\circ\text{C}$ for 2h before the reaction.

5. The chemical structures in Table 2, 3 and in other places both in the manuscript and supporting information are not properly drawn. Chemical structures should be corrected with proper orientation and bond length.

Answer: Done. (See revised **Tables 2 and 3** and **Figure S13**).

6. In table 3 it was given Ru: furfuraldehyde = 1: 400 or 1:200; it should be in general substrate instead of furfuraldehyde. What does it mean, Ru: furfuraldehyde = 1: 400 or 1:200? is this ratio related to weight of substrate or catalyst amount (or Ru content) or mmol of substrate: mmol of catalyst (Ru-content)? To be clear, it would be better to give the weight of catalyst used for the reaction and the mol% of Ru.

Answer: In the revised manuscript, the notes below Table 3 have been corrected and both the weight of catalyst and the mol% of Ru have been given.

7. The imidazole-based product 3a, was observed in case of furfuraldehyde or commonly this type of product forms in other aldehydes too? It would be better to check in few aldehydes also.

Answer: Based on the GC analysis results, it is found that the imidazole-based product forms only when the substrates are furfural, benzaldehyde and phenylacetaldehyde and the yields are all lower than 7% in these cases.

8. In the reductive amination of furfuraldehyde (Table 2), formation of desired product and only side product 3a is mentioned. But there should be the formation of secondary imine, secondary amine and corresponding alcohol. This needs to be checked properly and these products should be included in the reaction scheme and the yields in the table.

Answer: Based on the GC-MS analysis result for the reductive amination of furfural, the side products include only 3a and oligomers. The compounds that the reviewer mentioned were not detected.

9. This work is quite interesting, but authors need to demonstrate the applicability of their catalyst in more general sense by performing additional/broad substrate scope, and other experiments suggested above. Hence this manuscript requires suitable revision.

Answer: We again thank the reviewer's constructive suggestions, and based on

them we have performed additional experiments to address all the concerns of the reviewer. We hope this revised version is suitable for publication.

Reviewer #3

1. The authors report on a Ru single metal atom catalyst for the reductive amination. With regard to the reductive amination, the synthesis of primary amines, a challenging but highly interesting reaction was investigated. A rather low metal loading could be used but reaction conditions such as pressure of H₂ and temperature are relatively high. The addressed scope is interesting but the demonstrated tolerance of functional groups seems rather limited, may be at this stage. Single metal atom catalysis is meanwhile a rather mature field of research and catalyst development should lead to catalysts with an outstanding performance. Note, the catalyst synthesis strategy is not new (reference 36). Unfortunately, the authors missed to describe the state of the art comprehensively. The most comprehensive review (Chem. Rev. 2020) is not cited as recent development in the field of the synthesis of primary amines via reductive amination. At this stage, I can only weakly support publication and hope to see a revised version with a comprehensive discussion of the state of the art and a more impressive tolerance of functional groups.

Answer: We appreciate the reviewer's critical concerns on the single atom catalysis as well as the substrate scope. Herein we would like to address these concerns as below:

- a) **Regarding the single-atom catalysis (SAC):** we respectfully don't agree on the reviewer's comment that "Single metal atom catalysis is meanwhile a rather mature field of research and catalyst development should lead to catalysts with an outstanding performance". In fact, single-atom catalysis is far from mature although this field has been intensively studied in the past ten years since we first proposed this concept in 2011 (*Nat. Chem.*, 2011, **3**, 634-641), for example, it is yet unknown how far SAC can go (e.g., the scope of reactions that SAC can work better than nanocatalysts), how it works in a reaction (the reaction mechanism of SAC), how the coordination environment

of single atoms affects the catalysis, as well as can it be superior to nanocatalysts in terms of durability? To address those concerns will be of pivotal importance to the development of SAC and eventually guide the rational synthesis of efficient catalysts for practical applications. Precisely motivated by these open questions in SAC, in the present work we explore the Ru₁/NC SACs for application to a challenging reaction, the reductive amination of aldehydes/ketones to primary amines. To our knowledge, SACs have never been explored for this type of reactions before our present work although numerous M-N-C SACs have been synthesized for electrocatalysis. Through delicately designed characterizations, we are now able to address some of the key questions of SACs, see below.

- b) **Regarding the novelty:** we acknowledge that the catalyst synthesis method is not new, but this is not the focus of the present work. As we have demonstrated in the manuscript, the present work focus on the exploration of SACs for the more demanding and challenging reactions such as the reductive amination of aldehydes/ketones. The new findings in this work include 1) the superior activity and selectivity of Ru₁/NC SAC to the Ru nanocatalysts, and the highest thus far production rate of FAM per mass of Ru per hour has been achieved with our Ru₁/NC SAC; 2) the unrivalled robustness against being poisoned by CO, sulfur, and harsh hydrogen treatment is for the first time demonstrated, which cannot be accomplished by using previous Ru nanocatalysts; 3) we successfully address how the coordination structure of RuN_x affects the catalytic activity, which will provide guidelines for the rational design of efficient catalysts by fine-tuning the local structure of single atoms. All these new findings will help to address the key scientific issues in the field of SACs as we have mentioned in a).
- c) **Regarding the limit of the substrate scope:** we have extended the original substrate scope to more challenging and structurally diverse carbonyl compounds (15 newly added examples), and these new results are listed in the updated **Table 3**. As you see, for all these aldehydes/ketones substrates with

diverse structures and functional groups, our Ru₁/NC-900-800NH₃ single-atom catalyst worked well and afforded moderate to good yields, demonstrating the general applicability of the Ru₁/NC single-atom catalyst.

2: The most comprehensive review (Chem. Rev. 2020) is not cited as recent development in the field of the synthesis of primary amines via reductive amination.

Answer: We are sorry for missing this excellent review article, and now it has been cited in the revised manuscript (**Ref. 14**).

3. Scheme 1 is overdone! Many figures contain very small numbers and/or letters and need to get redesigned.

Answer: We are sorry for the unclarity of some figures. In the revised manuscript, we re-designed these figures so that they are presented in a more clear manner.

Scheme 1 is important for readers to understand the complexity of the reductive amination of aldehydes. All of the products and byproducts presented in scheme 1 have been reported in literature or in our present work depending on the catalyst nature. Therefore, we don't think it is overdone and prefer to maintain it unchanged.

REVIEWERS' COMMENTS

Reviewer #1 (Remarks to the Author):

The authors fully replied to my comments, and I believe that this paper is a step away from publication. The authors should compare the heterogeneous Ru catalysts presented in this paper with reported homogeneous Ru catalyst in TOF before publication. The reader would want to know whether there is difference in TOF between both catalytic systems, and what the difference is due to.

Reviewer #2 (Remarks to the Author):

Authors have addressed the comments and suitably revised the manuscript. Now this revised manuscript can be accepted for publication.

Reviewer #3 (Remarks to the Author):

The authors improved as suggested by me; in consequence, I can support publication now!

Responses to the referees

We are very grateful to the three reviewers for their careful reading of the revised manuscript and gave a positive evaluation. The point-by-point response to the reviewers are shown below.

Reviewer # 1:

1. The authors fully replied to my comments, and I believe that this paper is a step away from publication. The authors should compare the heterogeneous Ru catalysts presented in this paper with reported homogeneous Ru catalyst in TOF before publication. The reader would want to know whether there is difference in TOF between both catalytic systems, and what the difference is due to.

Answer: According to the reviewer's suggestion, we compared our best-performance Ru₁/NC-900-800NH₃ catalyst with more reported homogeneous Ru catalysts. As shown in Table S3, all the homogeneous catalysts (entry 17-23), irrespective of ligands, afforded much lower TOFs (2 ~ 20 h⁻¹) than our single-atom catalyst Ru₁/NC-900-800NH₃ which afforded a TOF of 176 h⁻¹ at milder conditions (100 °C, 2 MPa H₂), which is at least one order of magnitude higher than those of homogeneous catalysts. The low efficiency of homogeneous Ru catalysts might be due to the formation of stable Werner-type ammine complexes and the difficulty in activation of NH₃ (Beller, et al., *Nat. Commun.* 2018, 9, 4123). Compared with homogeneous Ru catalysts, our single-atom Ru catalyst is able to activate both NH₃ and H₂ which can be attributed to the low-coordinated and electron-rich Ru sites.

In the revised manuscript, the TOFs of both heterogeneous and homogeneous Ru catalysts have been added to Table S3, and the related text has been revised as:

“These strictly control experiments, together with the comparison with more heterogeneous and homogeneous Ru catalysts reported in literature (Table S3), unequivocally demonstrate the remarkable advantage of Ru₁/NC-900-800NH₃ SAC in maximizing the atom efficiency of Ru.”

Table S3. The superior performance of Ru₁/NC-900-800NH₃ as compared to the reported Ru and Rh nanocatalysts.^a

Entry	Catalyst	Molar ratio Ru/Substrat e	P _{NH₃} (MPa)	P _{H₂} (MPa)	Time (h)	Tem. (°C)	Amine yield (%)	Production Rate (g _{FAM} ·g _{Ru} ⁻¹ ·h ⁻¹)/ TOF (h ⁻¹)	Ref.
1	Ru ₁ /NC-900-800NH ₃	1:400	0.5	2	4	100	97	170.7 / 176	This work
2 ^b	Ru/Nb ₂ O ₅	1:400	0.5	2	6	100	91	115 / 119	This work
3 ^c	Ru/HZSM-5	1:400	0.5	2	7.5	100	68	89.6 / 92	This work
4 ^d	Ru ₁ /NC/Nb ₂ O ₅	1:400	0.5	2	3	100	97	227.6 / 235	This work
5	Ru/Nb ₂ O ₅	1:250	0.1	4	2	90	99	124.2	1
6	Ru nanoparticles	1:250	8 mmol	2	1.5	90	99	125.2	2
7	Ru/TiP	1:500	0.3	1.7	24	30	91	18.2	3
8	Ru/HZSM-5	1:27	7 mmol	3	0.25	100	76	103.5	4
9	Ru/Nb ₂ O ₅ -L	1:1000	10 mmol	2	8	90	60	72.8	5
10	Ru/Nb ₂ O ₅ -nH ₂ O-300	1:250	8 mmol	4	4	70	89	54.6	6
11	Ru/SiO ₂	1:670	1mL NH ₄ OH	7.5	4	130	90	145.5	7
12	Ru/γ-Al ₂ O ₃	1:266	0.4	3	2	80	75	96	8
13	Ru/ZrO ₂	1:40	29 mmol	2	12	85	94	1.9	9
14 ^e	Ru/C	1:114	17.6mmol	1.5	14	100	95	13.8	10
15 ^f	Ru/C	1:100	0.4	0.9	2	25	87.8	42.2	11
16 ^g	Rh/Al ₂ O ₃	1:2000	70mmol	2	2	80	91.5	45.8	12
17	RuCl ₂ (PPh ₃) ₃	1:50	0.5-0.7	4	24	130	85	2 / 2.1	13
18 ^h	Ru/amino acid/diphosphine	1:400	0.5-0.7	5	18	130	94	22.2 / 20.9	14
19 ⁱ	Ru(PPh ₃) ₃ H(CO)Cl + (S,S)-f-binaphane	1:100	2.5 equiv. NH ₄ I	3	24	80	84	4.4 / 3.5	15
20 ^j	Ru[(s)-BINAP](OAc)) ₂	1:100	5 equiv. NH ₄ OAc	5	24	80	94	7.7 / 3.9	16
21 ^k	Ru(Cl)H(CO)(PPh ₃) ₃	1:100	0.6	4	16	120	99	7.4 / 6.2	17
22 ^k	Trans-[Ru(NH ₃) ₂ {PP h ₂ (2-OC ₆ H ₄) ₂ }]	1:100	NH ₄ OAc	NaBH ₄	48	90	100	2.5 / 2.1	18
23 ^k	Ru(Cl)H(CO)(PPh ₃) ₃	1:100	0.6	4	16	120	99	7.4 / 6.2	19

^a Unless otherwise noted, the substrate was furfural. ^b The catalyst was prepared based on the

procedure reported in *J. Am. Chem. Soc.* 2017, **139**, 11493-11499; ^c The catalyst was prepared based on the procedure reported in *Mol. Cat.* 2020, **482**, 110755; ^d The catalyst was prepared with the similar procedure to that for Ru₁/NC-900-800NH₃ catalyst except for the employment of Nb₂O₅ support. ^e Glycolaldehyde as substrate; ^f (E)-1-(furan-2-yl)-5-methylhex-1-en-3-one as substrate; ^g Cyclohexanone as substrate; ^h Benzaldehyde as substrate; ⁱ Cyclohexyl methyl ketone as substrate; ^j Ortho-hydroxy-substituted diphenyl ketone as substrate; ^k Acetophenone as substrate.

Reviewer #2

1. Authors have addressed the comments and suitably revised the manuscript. Now this revised manuscript can be accepted for publication.

Answer: We appreciate very much for the reviewer's nice comment.

Reviewer #3

1. The authors improved as suggested by me; in consequence, I can support publication now!

Answer: We appreciate very much for the reviewer's nice comment.